# Effects of expanding imports on urban manufacturing employment: Evidence from China

**Shiping Wang**![ORCID], **Chunyan Zhao**![ORCID]*

International Business and Management Research Centre, Beijing Normal University, Zhuhai, China

* zhaochunyan@bnu.edu.cn

**Data Availability Statement:** All relevant data are within the manuscript and its Supporting information files.

**Funding:** This work is supported by One Belt and One Road College of Beijing Normal University,

## Abstract

Full employment is important to promote the high-quality development of the urban economy. Using urban-level data on China from 2004 to 2018, we analyse the effects and mechanism of expanding imports on urban manufacturing employment. We use the Guiding Opinions on Strengthening Import to Promote Balanced Development of Foreign Trade issued by the China State Council in 2012 as a natural experiment to solve the endogeneity problem. We find that expanding imports significantly increases urban manufacturing employment. This conclusion is still robust after a series of robustness tests. Further mechanism tests reveal that productivity improvements and upgrades to product quality from expanding imports can explain increased urban manufacturing employment. The results of the heterogeneity analysis show that expanding imports promote manufacturing employment in large and medium-sized cities but not small cities. Expanding imports increases employment in manufacturing in cities in different regions, with the largest effects on eastern cities, the second largest effects on western cities, and the smallest effects on central cities. These results suggest that expanding imports is an effective channel for increasing employment.

## 1 Introduction

Expanding imports is one of the major measures taken by China to promote high-quality development and is a long-term strategy of its opening up. Expanding imports is also an important way to meet people's needs for a better life. After the first explicit proposal of "actively expanding imports" at the Central Economic Work Conference in 2006, China has always attached great importance to expanding imports. Especially since the 18th National Congress of the Communist Party, China has placed unprecedented importance on actively expanding imports. The implementation of a series of import expansion policies has significantly promoted the development of China's imports. The import trade volume jumped from 11.48 trillion RMB in 2012 to 17.36 trillion RMB in 2021, with an average annual growth rate of 4.7%.

With the development of import trade, the analysis of import welfare effects has become a hot topic in the academic field. As the most populous developing country, promoting stable

Zhuhai, China (Grant No. 2019BRSKYC002), the 2022 Characteristic Innovation Project of the Ordinary Colleges and Universities of the Department of Education of Guangdong Province (Grant No. 2022 WTSCX181), and 2023 Discipline Co-construction Project of Guangdong Provincial Philosophy and Social Sciences (Grant No. GD23XYJ02). The funders provide us with financial aid, which can be used for participation in academic conferences, collection of data, and publication of manuscript. This manuscript is one of the phase achievements of the funded projects. There was no additional external funding received for this study.

**Competing interests:** The authors have declared that no competing interests exist.

employment growth has always been an important issue related to China's economic development and even social stability. Therefore, the question that arises is, what exactly is the impact on employment of expanding imports? The literature has explored this issue from different perspectives. Given the increasing availability of micro data at the firm level, recent research has mainly analysed the impact of firms' import behaviour on employment. However, scant literature has focused on the urban level. As an important spatial node in a country's foreign trade network system, cities are the main force in promoting the high-quality development of foreign trade. *The 14th Five Year Plan* adopts the promotion of the high-quality development of the urban economy and the urbanization of the agricultural transfer population as the core issue of China's economic construction and an important component of building a modern economic system. Therefore, studying the effects of increasing imports on urban employment provides theoretical and empirical evidence for formulating policies to accelerate high-quality, full employment.

Preliminary observations of the data samples in this paper reveal an overall upwards trend and positive correlation between urban imports and manufacturing employment (see Figs 1 and 2). Has expanding imports promoted urban manufacturing employment? If so, what is the mechanism? Is there a difference due to the heterogeneity of cities' locations and sizes?

To address these questions, we start by constructing a matched database using the China Urban Statistical Yearbook and China Regional Economic Statistical Yearbook for 2005 to 2019. We use *Guiding Opinions on Strengthening Imports to Promote Balanced Development of Foreign Trade* (hereinafter referred to as the *Guiding Opinions*) issued by the China State Council in 2012 as a natural experiment of expanding imports and the difference-in-differences (DID) strategy to compare urban manufacturing employment in the pre- and post policy periods to determine whether expanding imports increases urban manufacturing employment. We find that when imports increase by 1%, urban manufacturing employment increases by 1.9%. In particular, this effect is driven mainly by the productivity increase and product quality upgrading resulting from expanding imports. To test whether the baseline findings are reliable, we conduct a robustness test and still find a positive and significant effect of expanding imports on urban manufacturing employment.

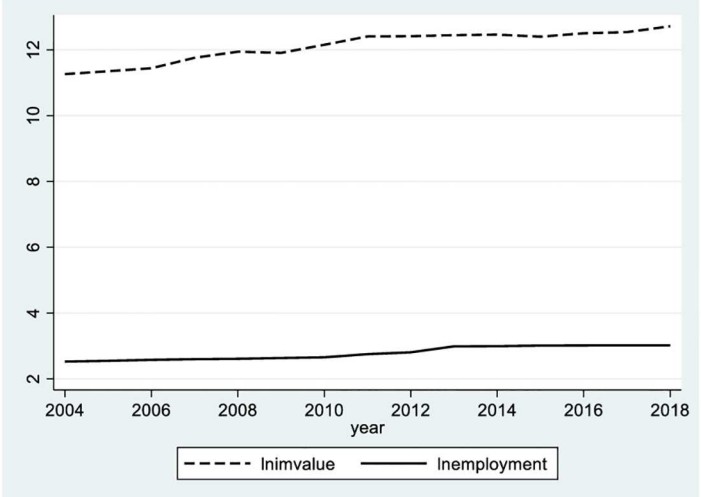

**Fig 1. Trends in urban import and manufacturing employment.** The dashed line represents the logarithm of the annual urban import volume, the solid line represents the logarithm of the annual urban manufacturing employment scale.

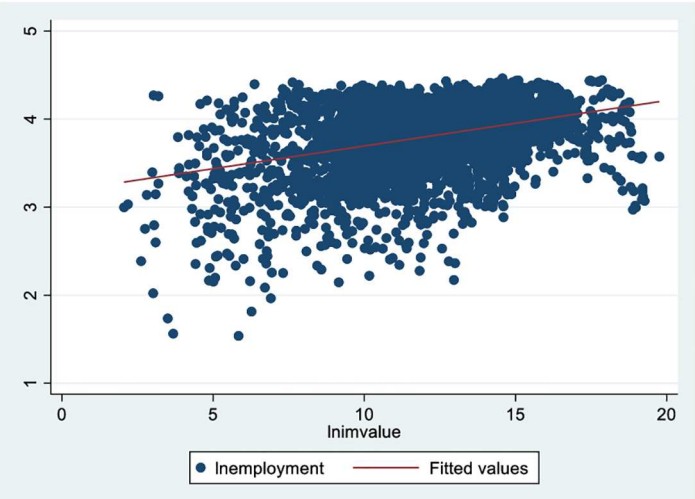

**Fig 2. Correlation between urban imports and manufacturing employment.** The point represents the logarithmic value of the annual employment scale of 285 sample cities from 2004 to 2018.

After establishing a causal link between expanding imports and urban manufacturing employment, we continue to examine heterogeneous responses in terms of cities' locations and sizes. We find that expanding imports promote manufacturing employment in large and medium-sized cities but not small cities. Expanding imports encourages manufacturing employment in cities in different regions, with the largest effects on eastern cities, the second largest effects on western cities, and the smallest effects on central cities.

These findings complement those of Hayakawa et al. (2021), who document the positive impact of import penetration in the industries from which firms purchase their inputs [1]. These findings also relate to Lai et al. (2022), who argue that industries facing more intense import competition have lower workplace injuries and illness rates [2]. While these results suggest the positive impacts of imports on employment from the perspective of industries or firms, they do not yet reflect the exploitation of the effects of imports on urban manufacturing employment. In this paper, we further deepen the understanding of the welfare effects of expanding imports and provide theoretical and empirical evidence at the urban level to achieve the goal of accelerating the urbanization of the agricultural transfer population and promoting high-quality and full employment.

The rest of this paper is arranged as follows. The second part reviews the related literature and presents the paper's contribution. The third part introduces the background to import expansion policies. The fourth part describes the data, model and empirical strategy. The fifth part presents and interprets the baseline empirical findings. The sixth part further tests the potential channels through which expanding imports increase urban manufacturing employment. The seventh part presents the heterogeneity test. The last part presents the conclusion.

## 2 Related literature and contribution

The effects of international trade on employment have always been the focus of international economics. Since the 1980s, many studies have begun to focus on the impact of export trade on employment [3]. Given the gradual deepening of the global industrial and value chain division of labour, the effects of imports on employment have gradually received increasing attention [4].

## 2.1 Imports promote employment

The relevant literature has suggested that imports are one of the main factors promoting employment. Imports promote the growth of domestic output through the trade multiplier effect, thereby increasing the demand for labour and promoting an expansion in employment. Importing advanced capital goods, especially advanced machinery and equipment, can not only directly increase the employment of highly skilled labour but also improve the employment structure and promote labour employment growth by updating production equipment and improving production efficiency. From the perspective of consumer goods, although most consumer goods imports do not have a creative effect on domestic employment, the consumption demonstration effect and import competition effect generated by consumer goods imports stimulate domestic firms to increase R&D investments and imitate the production of imported new products, which encourages firms' innovation and increases their price markups [5], and employment growth is significantly promoted. From the perspective of intermediate goods, the import learning effect brought by intermediate goods imports has a significant positive impact on the micro performance of firms, such as R&D and production efficiency and increases the demand for skilled labour in China [6, 7]. Shao et al. (2019) study the impact of expanding imports on employment from the perspective of resource allocation. They find that expanding imports reduces the distortion of production factor allocation caused by trade barriers, optimizes production factor allocation, and thus improves economic growth efficiency and potential, with a longer-term promoting effect on employment [8]. Xu and Wang (2020) note that the technology spillover effect from expanding imports promotes the technological progress of domestic firms and reduces production costs and entry barriers, thereby attracting more firms to enter the market and creating more job opportunities [9]. Bloom et al. (2019) show that import competition increases employment in the service sector in areas with high human capital and in large firms [10]. Jongwanich et al. (2022) find that in Thailand, there is a diminished negative impact resulting from imports [11].

## 2.2 Imports inhibit employment

Some studies have found that imports have an inhibitory effect on employment. Chen et al. (2017) argue that an increase in intermediate input imports reduces the employment scale [12]. The same conclusion was reached by Rodriguez-Lopez and Yu (2017) [13]. Ram and Acharya (2017) use data from 88 Canadian industries from 1992 to 2007 and find that import growth inhibits employment [14]. Autor et al. (2016) also find that expanding imports had a significant inhibitory effect on employment. A large body of the literature has explored how import competition affects employment [15]. Balsvik et al. (2015) study the impact of Chinese import competition on the labour market in Norway and find that imports from China reduced employment in the Norwegian manufacturing industry, especially for low-skilled workers [16]. Autor et al. (2014) examine the impact of imports on the US labour market within 722 commuting zones. Their findings show that import competition led to a noticeable decline in manufacturing employment across all major occupation groups [17]. Acemoglu et al. (2016) seek to explain employment losses at the industry level [18]. Their central estimates suggest job losses from rising Chinese import competition in the 2.0–2.4 million range from 1999 to 2011. Adda and Fawaz (2020) find that in regions with higher proportions of jobs involving routine tasks, import competition discourages labour market participation [19]. Wei and Lian (2020) document that import competition from downstream industries decreases the demand for domestic intermediate goods in downstream industries, thereby suppressing employment growth [20].

## 2.3 Contribution of the paper

In this paper, we contribute to the literature in several ways. First, unlike most studies that have focused on the industry or firm level, we study the effects of expanding imports on employment from the perspective of cities. Furthermore, we characterize and test the mechanisms by which expanding imports impact urban manufacturing employment. Additionally, we distinguish urban size and location to analyse the differential impact of expanding imports on urban manufacturing employment. Most importantly, we take the *Guiding Opinions* issued by the China State Council in 2012 as an exogenous policy shock. Within the framework of natural experiments, a multiplier model is constructed to effectively solve the endogeneity problem and identify the causal effect between expanding imports and urban employment.

In summary, we further deepen the study of the economic effects of expanding imports and provide reliable and robust empirical evidence at the urban level to implement expanding import policies, accelerate the urbanization of the agricultural transfer population and promote high-quality and full employment.

## 3 Institutional background of enlarging import policy

The process of evolution of China's import policy can be divided into four approximate stages: the initial stage; the stage in which exports are the main focus and imports are the auxiliary focus; the stage in which imports and exports are equally important; and the stage of active expansion imports. After its accession to the WTO in 2001, China's opening up to the outside world entered a new stage of the rapid development of import trade. The Central Economic Work Conference, held on December 14, 2011, explicitly proposed to "strengthen and improve import work, actively expand imports, and promote trade balance". Starting on January 1, 2012, China implemented lower temporary import tax rates on more than 730 commodities. The Foreign Trade Department of the Ministry of Commerce also stated that during the 12th Five Year Plan period, China will take further measures to expand imports. On April 30, 2012, the State Council issued the *Guiding Opinions*, proposing that "while maintaining stable export growth, we should pay more attention to imports, appropriately expand import scale, promote balance of foreign trade, and achieve sustainable development of foreign trade" (see Table 1 for detailed and specific measures). Premier Li Keqiang stressed in the 2018 Report on the Work

**Table 1. Guiding opinions on the main specific measures for expanding imports.**

| Strategies | Specific measures |
| --- | --- |
| Adjust import tariffs on some goods | Reduce import tariffs on some energy raw materials; appropriately reduce import tariffs on some daily necessities closely related to people's lives; adjust import tariffs on some advanced technology equipment and key components; and focus on reducing import tariffs on key components that cannot be produced domestically or whose performance does not meet the needs of primary energy raw materials and strategic emerging industries. |
| Strengthen and improve financial services | Provide diversified financing facilities; improve import credit insurance system and trade settlement system. |
| Improve management measures | Further optimize the management of import processes; improve the import management of special customs supervision and bonded supervision areas; promote the connection between imports and domestic circulation; promote the transformation and upgrading of processing trade; improve the early warning mechanism for industrial damage; improve quality and safety warning mechanism for imported goods. |
| Improve trade facilitation levels | Further improve customs clearance efficiency; strengthen the infrastructure construction of border trade; strengthen the construction of e-government information platforms. |

of the Government that it is necessary to "actively expand imports", reduce import tariffs on cars and consumer goods, and promote industrial upgrading and balanced trade development. The Ministry of Commerce, together with 20 departments, such as the Development and Reform Commission and the Ministry of Finance, has drafted and formed the "Opinions on Expanding Imports to Promote the Balanced Development of Foreign Trade", which propose the full leveraging of the important role of imports in improving consumption, adjusting the structure, developing the economy, and expanding openness while stabilizing the international market share of exports and encouraging the balanced development of imports and exports. The document officially emphasizes the importance of "actively expanding imports" in trade policies. In January 2019, the Ministry of Commerce and the General Administration of Customs reiterated at a press conference that imports will be actively expanded, and China will successfully hold the Second China International Import Expo. In December 2019, the State Council Information Office of the People's Republic of China proposed four measures to expand imports at a press conference: reduce tariffs, organize import expos, improve the level of trade facilitation with high standards, and cultivate a batch of demonstration zones to promote innovations in import trade. In January 2020, the regular press conference of the Ministry of Commerce proposed to cultivating numerous import trade promotion demonstration zones, promoting the construction of import trade innovation platforms with points and areas by cultivating import trade distribution centres, and constantly improving the radiation driving role of demonstration zones.

A series of import expansion policies has significantly promoted the development of China's import trade. Considering data availability, we use the *Guiding Opinions* released in 2012 as a natural experiment to study the effects of expanding imports on urban employment.

## 4 Mechanism analysis

We hold that urban imports may expand manufacturing employment through productivity improvement effects and product quality upgrading effects.

### 4.1 Productivity improvement

The literature has confirmed the productivity improvement effect of imports. Amiti and Koning (2007) study the impact of tariff reductions on the productivity of Indonesian firms and find that a 10% reduction in import tariffs can increase firm productivity by 3%. For firms importing intermediate goods, productivity increases can reach 11% [21]. Mo et al. (2021) study the distinctive productivity-improving effects of intermediate imports and capital goods imports and find a substantially larger productivity effect caused by capital goods imports [22]. The studies by Kasahara and Rodrigue (2008) on Chile, Halpern and Koren (2007) on Hungary and Li et al. (2023) on China also provide evidence of the positive relationship between imports and productivity [23–25]. Productivity improvement is an important way for imports to promote domestic employment growth [26]. The import of new products and intermediates containing advanced technology can effectively improve firm performance, promote technological progress and total factor productivity, and stimulate greater production capacity. As a result, the demand for highly skilled labour also increases, thereby promoting urban employment [27]. In addition, from a dynamic perspective, expanding imports can promote the optimization of factor allocation and technology spillover, promote domestic technological progress in importing countries, and improve production efficiency, thereby forming a more long-term, promoting effect on employment. Beaudry et al. (2006), Bartel et al. (2007) and Graetz and Michaels (2018) focus on outcomes from technological advancements and show

that, to some extent, technological advancements improve labour productivity and thus employment [28–30].

### 4.2 Product quality upgrade

The quality upgrading effect generated by expanding imports is another channel encouraging urban employment. Intermediate goods imports have significantly improved export product quality through technology spillover effects [31]. Maria Bas and Vanessa Strauss-Kahn (2015) suggest that firms exploit input trade liberalization to upgrade the quality of their inputs and, thus, their exported products [32]. Fan et al. (2015) show that an increase in imported intermediates improves export quality [33]. Quality improvements thereby increase the international competitiveness of export firms, enhancing the stability of exports, expanding the export market share, and ultimately increasing urban manufacturing employment. Bas and Paunov (2021) emphasize the complementarity between high-quality imported inputs and skilled labour, which jointly improves output quality [34]. From the perspective of productive service imports, expanding imports enables manufacturing products and their quality to be upgraded and improved through complementary effects, technology spillover effects, competitive effects, and specialized division of labour effects [5], improving the possibility for increasing marginal returns in production and, thus, creating more employment opportunities for highly skilled labour, thereby promoting employment growth in urban manufacturing. From the perspective of digital trade, the import of digital products has significantly improved the quality of high-end manufacturing products, promoting further expansions in exports and employment.

## 5 Research design

### 5.1 Model

We construct the following econometric model to test whether expanding imports promotes urban manufacturing employment.

$$Y_{ct} = \alpha + \beta import_{ct} + \gamma X_{ct} + \mu_c + \mu_t + \varepsilon_{ct} \tag{1}$$

$Y_{ct}$ represents the logarithmic value of the manufacturing employment level of urban $c$ in year t, $import_{ct}$ represents the logarithmic value of the import volume of urban $c$ in year $t$, $X_{ct}$ represents the set of control variables, $\mu_c$ and $\mu_t$ are used to control urban and year fixed effects, respectively, and $\varepsilon_{ct}$ is a random error term.

To ensure that the core variable $import_{ct}$ in Model (1) is an unbiased estimator, it needs to satisfy a crucial assumption: after controlling for all control variables, $import_{ct}$ and the error term $\varepsilon_{ct}$ become irrelevant. However, it is possible that cities with large import volumes have substantial economic development and a strong attraction for labour, which leads to biased estimations of the impacts of expanding imports on urban employment. To solve the endogeneity problem, we use the *Guiding Opinions* issued by the China State Council in 2012 as a tool variable for expanding imports.

We set up a treated and a control group based on the trade balance situation (i.e., export amount minus import amount). Specifically, if an urban area experiences a foreign trade deficit, it is designated as the treated group; otherwise, it is set as the control group. Then, we study the changes in the employment scale between the treated group and the control group before and after the issuance of the Guiding Opinions. Using the above research ideas, referring to Bao and Huang (2022) [35], we construct the following multiplier model.

$$lnEmp_{ct} = \alpha + \beta treat_c \times Post_{12} + \gamma X_{ct} + \mu_c + \mu_t + \varepsilon_{ct} \tag{2}$$

The explained variable $lnEmp_{ct}$ represents the employment level of urban $c$ in year $t$, and $treat_c \times Post_{12}$ is the explanatory variable.

## 5.2 Data source

The data used in this paper mainly come from the China Urban Statistical Yearbook and China Regional Economic Statistical Yearbook. They are informative annual publications organized by the National Bureau of Statistics, that comprehensively reflect the economic and social development of Chinese cities. All relevant indicators used in this paper, such as urban employment level, urban economic agglomeration level, and urban income level, are the statistical indicators of the "municipal area" in the China Urban Statistical Yearbook rather than the statistical indicators of the "whole urban area"; that is, when using the statistical indicators of the "municipal area", the relevant statistical data of counties, county-level cities and administrative regions below the urban jurisdiction are excluded to more accurately depict the economic and trade development status of the urban area itself. Considering that the China Urban Statistical Yearbook has increased the industrial categories from 15 to 19 since 2004, we use its relevant data and data from the China Regional Economic Statistical Yearbook during 2005–2019 and finally obtain 4273 observation samples of 285 cities at the prefecture level and above in China from 2004 to 2018.

## 5.3 Variable selection and description

**(1) Explained variable.** Urban manufacturing employment ($lnEmp_{ct}$): defined as the proportion of the number of employees in the secondary industry to the number of employees in the urban area at the end of the year.

**(2) Explanatory variable.** $treat_c \times Post_{12}$: $treat_c$ represents the dummy variable. As mentioned earlier, we first measure and calculate the trade balance (i.e., export value minus import value) of 285 cities in 2012 to set up a treated group and a control group. If urban foreign trade is in a deficit, $treat_c = 1$; otherwise, $treat_c = 0$. $Post_{12}$ represents the dummy variable of import expansion policies. If the data year is 2012 or later, $Post_{12} = 1$; otherwise, $Post_{12} = 0$.

**(3) Control variable**.

① Economic agglomeration level (lnagglomeration). This variable is defined as the proportion of the number of industrial firms to the land area (square kilometres) in cities.

② Urban income level (lnwage). This variable is defined as the average urban wage.

③ Urban consumption level (lnconsumption). This variable is defined as the value of the total retail sales of social consumer goods in cities.

④ Urban supply level (lnsupply). This variable is defined as the total output value of industrial firms.

**(4) Mediating variable.** We use the level of urban productivity and the complexity of urban export technology as mediating variables to test the mechanism by which expanding imports impacts urban employment.

① Urban productivity (lnTFP). We use the Malmquist index with 2004 as the base period to calculate the urban TFP through cumulative multiplication. The actual GDP of each urban area is used to represent the output variable, and the GDP index of each province where the urban area is located is used as the base period for adjustments in 2004. Employment numbers and capital stock are used to represent input variables.

② Complexity level of urban export technology (lnESI). We first calculate the complexity of export technology at the product level.

$$PRODY_h = \sum_{m=1}^{n} \frac{\left(EXP_{m,h}/EXP_m\right) \times Y_m}{\sum_{m=1}^{n}\left(EXP_{m,h}/EXP_m\right)} \tag{3}$$

$EXP_{m,h}$ represents the export amount of product $h$ in country m, $EXP_m$ represents the total export amount of country m, $EXP_{m,h}/EXP_m$ represents the proportion of the export trade volume of product $h$ in the total export volume of country m, $Y_m$ represents the productivity of country m, and per capita GDP is used as a proxy for productivity. After obtaining the export technology complexity at the product level through the above method, it is weighted by the export trade volume of the product and added up to the urban level to obtain the urban export technology complexity ($ESI_{ct}$):

$$ESI_{ct} = \sum_{h=1}^{n} \frac{EXP_{c,h,t}}{\sum_{h=1}^{n}\left(EXP_{c,t}\right)} \times PRODY_h \tag{4}$$

$EXP_{c,h,t}$ represents the export quantity of product $h$ of urban $c$ in year $t$, and $EXP_{c,t}$ represents the total export quantity of urban $c$ in year $t$.

The descriptive statistics of variables are shown in Table 2.

## 6 Empirical results

### 6.1 Baseline estimation

We estimate the impact of expanding imports on urban manufacturing employment using Eq. (2). The results are presented in Table 3. Column (1) reflects the estimation results after controlling for urban and year fixed effects. It shows that the interaction term $treat_c \times Post_{12}$ is significantly positive at the 1% level, indicating that after the implementation of import expansion policies, the employment of the treated group's cities has significantly improved compared to the control group's cities. On the basis of the benchmark estimation, we add control variables and find that the estimated coefficient of $treat_c \times Post_{12}$ remains significantly positive, with imports increasing by 1% and urban employment increasing by 1.9%.

For the control variables, the impact of urban economic agglomeration on manufacturing employment is significantly positive at the 1% level. The main reason is that the thick labour market effect brought by urban economic agglomeration, as well as the close forwards and backwards correlation effects [36], can provide more job opportunities for labour with different skill structures and technical complexities and can also effectively improve employment matching efficiency and quality [37], thereby increasing employment in the urban

**Table 2. Descriptive statistics of variables.**

| Variable | Obs | Mean | Std. dev. | Min | Max |
|---|---|---|---|---|---|
| lnimvalue | 4,275 | 12.11338 | 2.677031 | 2.055238 | 19.75602 |
| lnemployment | 4,275 | 3.802299 | 0.3777412 | 1.536867 | 4.4628 |
| lnagglomeration | 4,275 | 0.2478976 | 0.2692348 | 0.0008553 | 1.653147 |
| lnconsumption | 4,275 | 16.55651 | 1.366743 | 6.866933 | 20.95981 |
| lnwage | 4,275 | 10.46363 | 0.5526933 | 7.58557 | 11.91734 |
| lnsupply | 4,275 | 15.38497 | 1.572724 | 8.973985 | 19.58426 |
| lnTFP | 4,275 | 0.3930674 | 0.696471 | -3.176677 | 1.482337 |
| lnESI | 4,275 | 9.461372 | 0.57871937 | 0.403545 | 10.93773 |

**Table 3. Baseline estimation results.**

| Variable | (1) | (2) |
|---|---|---|
| treat×Post$_{12}$ | 0.0186*** | 0.0190* |
| | (0.00691) | (0.00993) |
| lnagglomeration | | 0.0963*** |
| | | (0.0315) |
| lnconsumption | | 0.161*** |
| | | (0.00713) |
| lnwage | | 0.133*** |
| | | (0.0235) |
| lnsupply | | 0.150*** |
| | | (0.00807) |
| Constant | 3.797*** | 2.884*** |
| | (0.0195) | (0.269) |
| Urban fixed effect | YES | YES |
| Year fixed effect | YES | YES |
| Observations | 4,275 | 4,275 |
| R-squared | 0.094 | 0.107 |
| Number of urban | 285 | 285 |

Note:

\*\*\*, \*\*, and * denote significance at the 1%, 5%, and 10% levels, respectively. Standard errors are in parentheses.

manufacturing industry. The impact of the urban income level on employment is significantly positive at the 1% level. The main reason is that the labour supply curve is elastic compared to the labour demand curve due to the abundant labour force in China, which means that an increase in the labour income level significantly increases the labour supply. In addition, because the current income level is lower than the equilibrium income level, employment significantly increases with an increase in the income level, thereby promoting growth in manufacturing employment. The impact of the urban consumption level on employment is significantly positive. The main reason is that consumption is one of the three factors driving urban economic growth and an important force in promoting urban employment growth. In addition, increasing income and upgrading the consumption structure will also have a significant driving effect on employment. The impact of the urban supply level on employment is significantly positive, indicating that the expansion of production effectively improves manufacturing employment. The main reason is that good level of economic development is the foundation for sustained and healthy regional economic development (Li et al., 2022) [38]. Improvements in the urban supply level can expand production, thereby enabling employment in the manufacturing industry to increase.

## 6.2 Robustness test

**6.2.1 Parallel trend test.** An important prerequisite for using the DID method is the parallel trend hypothesis, which states that if the treated group is not affected by a policy intervention, its time trend should be the same as that of the control group. We conduct a parallel trend test to verify whether the parallel trend hypothesis can be satisfied. The results are shown in Fig 3. We see that before the State Council issued the *Guiding Opinions* in 2012, the regression estimation coefficient is relatively small, indicating that before the implementation of import expansion policies, the control and treated groups have the same development trend,

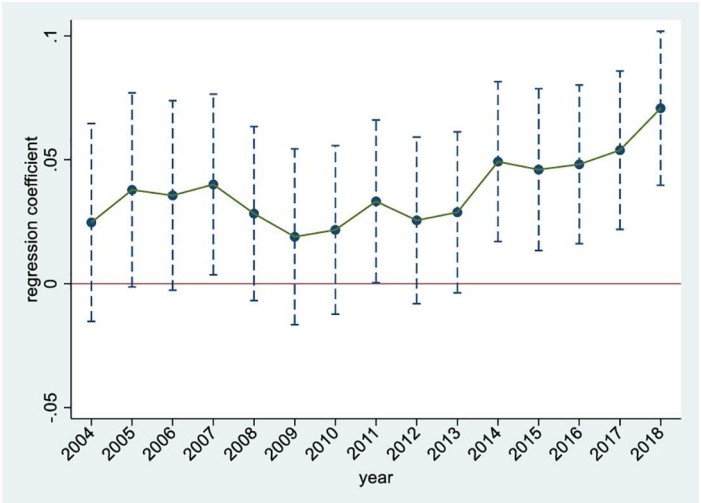

**Fig 3. Parallel trend test.** The dashed line represents the 95% confidence interval.

meeting the parallel trend assumption. From the perspective of dynamic effects, the estimated coefficient of employment in the treated group shows an overall upwards trend over time compared to that of the control group, indicating that the implementation of import expansion policies can effectively promote urban manufacturing employment.

**6.2.2 Expected effect test.** Due to the possibility that cities may anticipate the implementation of import expansion policies and alter trade behaviour decisions in advance, thereby undermining the parallel trend hypothesis, we conduct expected effect testing. The specific approach is to construct the time dummy variable $ProPost_{12}$ from the year before the issuance of the Guiding Opinions and then generate the interaction term $treat \times ProPost_{12}$ and include it in Model (2) for estimation. The estimated results are shown in Column (1) of Table 4. We find that the estimated coefficient of $treat \times ProPost_{12}$ is positive but not significant, indicating that before implementing import expansion policies, cities do not form in advance the expected effect of changing trade behaviour. In other words, implementing import expansion policies has strong exogenous effects.

**6.2.3 Placebo test.** To ensure that the benchmark estimation results are not affected by other unobservable factors, we conduct a placebo test by constructing a false implementation period for import expansion policies. The specific approach is as follows. First, the samples are removed from the implementation of import expansion policies in 2012 and subsequent years (i.e., 2012–2018). Second, during the sample period before the implementation of import expansion policies (i.e., 2004–2011), such implementations are advanced by three and four years, respectively, to construct dummy variables for the impact of the pseudo period of these policies. The dummy variables $Pre3$ and $Pre4$ are sequentially set, and the interaction term $treat$ is generated using the grouped dummy variables $treat \times Pre3$ and $treat \times Pre4$. Finally, the interaction term is included in Model (2) for estimation, and the results are shown in Columns (2) and (3) of Table 4. The coefficients of the interaction terms $treat \times Pre3$ and $treat \times Pre4$ are not significant, indicating that urban manufacturing employment is not affected by other unobservable factors.

**6.2.4 Two-stage DID estimation.** The significance of explanatory variables may be overestimated due to the possibility of sequence-related issues when using the multiperiod DID method. We re-estimate Model (2) using the two-period DID method. The specific approach

**Table 4. Robustness test.**

| Test method | (1) | (2) | (3) | (4) | (5) | (6) |
|---|---|---|---|---|---|---|
| | Expected effect test | Placebo test | Placebo test | Two stage DID estimation | Exclude cities with administrative changes | Remove outliers |
| Explained variable | lnEmp | lnEmp | lnEmp | lnEmp | lnEmp | lnEmp |
| treat×Post$_{12}$ | 0.0339** | | | 0.0193* | 0.0330*** | 0.0175** |
| | (0.0145) | | | (0.00995) | (0.0115) | (0.00768) |
| treat×ProPost$_{12}$ | 0.0157 | | | | | |
| | (0.0244) | | | | | |
| treat×Pre3 | | 0.00976 | | | | |
| | | (0.00678) | | | | |
| treat×Pre4 | | | 0.00962 | | | |
| | | | (0.00654) | | | |
| lnagglomeration | 0.0996*** | 0.0887** | 0.0847** | 0.0966*** | 0.0761* | 0.0787*** |
| | (0.0316) | (0.0366) | (0.0367) | (0.0316) | (0.0426) | (0.0245) |
| lnconsumption | 0.00467 | 0.0182 | 0.0178 | 0.00488 | 0.0137 | 0.0187* |
| | (0.0105) | (0.0139) | (0.0139) | (0.0105) | (0.0120) | (0.00961) |
| lnwage | 0.131*** | 0.0698*** | 0.0696*** | 0.133*** | 0.200*** | 0.0712*** |
| | (0.0235) | (0.0259) | (0.0259) | (0.0235) | (0.0277) | (0.0183) |
| lnsupply | 0.150*** | 0.0592*** | 0.0580*** | 0.150*** | 0.167*** | 0.107*** |
| | (0.00807) | (0.0111) | (0.0111) | (0.00807) | (0.00997) | (0.00658) |
| Constant | 2.869*** | 3.908*** | 3.918*** | 2.882*** | 3.174*** | 2.697*** |
| | (0.269) | (0.348) | (0.348) | (0.269) | (0.309) | (0.219) |
| Observations | 4,275 | 2,280 | 2,280 | 4,275 | 2,833 | 3,417 |
| R-squared | 0.108 | 0.045 | 0.045 | 0.107 | 0.134 | 0.125 |
| Number of urban | 285 | 285 | 285 | 285 | 189 | 275 |

Note:

***, **, and * denote significance at the 1%, 5%, and 10% levels, respectively. Standard errors are in parentheses.

is to use the *Guiding Opinions* issued by the State Council in 2012 to reflect the timeline. First, the sample is divided into two stages: 2004–2011 and 2012–2018. Second, we calculate the trade balance of cities in each stage and set cities with trade deficits as the treated group and cities with trade surpluses as the control group. Finally, we estimate Model (2) using the same approach as that used for the previous benchmark estimation. Column (4) of Table 4 shows that the estimated coefficient of *treat×Post$_{12}$* remains significantly positive, further indicating that the implementation of import expansion policies has a significant promoting effect on urban manufacturing employment.

**6.2.5 Exclude cities with administrative changes.** In recent years, China's urbanization has increased year by year. Given the acceleration of urbanization, some cities have also adjusted their administrative divisions. Such adjustments may affect the relationship between urban imports and manufacturing employment. Therefore, we exclude from the total sample cities with administrative divisions that have been adjusted during the study period and retest the impact of expanding imports on employment.

Through the National Administrative Division Information Query Platform, we find that 99 cities at the prefectural level and above adjusted their administrative divisions from 2004 to 2018[1]. We exclude the sample data of these 99 cities from the total sample and ultimately obtain 2833 estimated samples from 189 cities during 2004–2018. The results are shown in Column (5) of Table 4 and indicate that the estimated coefficient of *treat×Post$_{12}$* is significantly

positive at the 1% level; therefore, the implementation of import expansion policies has significantly promoted urban manufacturing employment, indicating that the benchmark estimation results are robust.

**6.2.6 Remove outliers.** The existence of outliers in the sample may have a certain impact on the estimation results. We re-estimate Model (2) after removing outliers. We first calculate the average value, 10 quantile and 90 quantile of urban employment; then, the samples with employment scales greater than the 90 quantile and less than the 10 quantile are removed from the estimated samples, and 3417 sample data points of 275 cities are obtained. Column (6) of Table 4 shows that the coefficient of $treat \times Post_{12}$ is significantly positive at the 5% level, indicating that the implementation of import expansion policies has a significant promoting effect on urban manufacturing employment. The benchmark estimation results are robust.

## 6.3 Mechanism test

In this section, we test the mechanism by which expanding imports impacts urban employment. We use the interaction variable $treat \times Post_{12} \times lnTFP$, which includes the grouping dummy variable $treat$, the policy implementation dummy variable $Post_{12}$, and the productivity improvement variable $lnTFP$ to test the productivity improvement mechanism. We use the interaction variable $treat \times Post_{12} \times lnESI$, which includes the group dummy variable $treat$, policy implementation dummy variable $Post_{12}$, and urban export technology complexity level $lnESI$, to test the quality upgrading mechanism.

The results of the productivity improvement mechanism test are shown in Columns (1) and (2) of Table 5, and the product quality upgrading mechanism test is shown in Columns (3) and (4). Columns (1) and (3) represent the estimated results without adding control

**Table 5. Mechanism test.**

| mechanism | (1) | (2) | (3) | (4) |
|---|---|---|---|---|
| | productivity increasing | productivity increasing | quality upgrading | quality upgrading |
| Explained variable | lnEmp | lnEmp | lnEmp | lnEmp |
| $treat \times Post_{12} \times lnTFP$ | 0.0127* | 0.0142** | | |
| | (0.00650) | (0.00711) | | |
| $treat \times Post_{12} \times lnESI$ | | | 0.00209*** | 0.00207* |
| | | | (0.000737) | (0.00106) |
| lnagglomeration | | 0.0898*** | | 0.0917*** |
| | | (0.0312) | | (0.0318) |
| lnconsumption | | 0.00503 | | 0.00314 |
| | | (0.0105) | | (0.0109) |
| lnwage | | 0.135*** | | 0.135*** |
| | | (0.0234) | | (0.0238) |
| lnsupply | | 0.150*** | | 0.151*** |
| | | (0.00807) | | (0.00815) |
| Constant | 3.801*** | 2.903*** | 3.797*** | 2.921*** |
| | (0.0197) | (0.268) | (0.0198) | (0.274) |
| Observations | 4,275 | 4,275 | 4,275 | 4,275 |
| R-squared | 0.054 | 0.108 | 0.091 | 0.108 |
| Number of urban | 285 | 285 | 281 | 281 |

Note:

***, **, and * denote significance at the 1%, 5%, and 10% levels, respectively. Standard errors are in parentheses.

variables, and Columns (2) and (4) represent the results after adding the control variables. The estimation results show that regardless of whether control variables are added, the interaction terms *treat×Post_{12}×lnTFP* and *treat×Post_{12}×lnESI* are significantly positive, indicating that expanding imports increases manufacturing employment by improving urban productivity and product quality.

## 6.4 Heterogeneity analysis

To examine whether the impact of imports on urban manufacturing employment varies depending on the location and size of the urban area, we categorize cities by size and location and conduct heterogeneity tests.

**6.4.1 Heterogeneity at the urban scale.**   According to the *Opinions of the State Council on Further Promoting the Reform of the Registered Residence System of China* issued on July 30, 2014, cities with a permanent population of more than 5 million in urban areas are megacities, cities with a permanent population of 1–5 million in urban areas are large cities, cities with a permanent population of 500–1000 thousand in urban areas are medium-sized cities, and cities with a permanent population of less than 500 thousand in urban areas are small cities. We collectively refer to megacities and large cities as large cities, while the classification criteria for medium-sized and small cities remain unchanged. We then estimate the impact of expanding imports on manufacturing employment in cities of different scales. The results are shown in Table 6. The implementation of import expansion policies has a significant positive impact on employment in large and medium-sized cities but a nonsignificant impact in small cities.

We believe that the main reason for this result is that the larger the urban size is, the higher the productivity and wage levels. The agglomeration and ranking effects accelerate the entry of more low-cost or high-productivity firms, as well as high-skilled labour, into larger cities.

**Table 6. Urban scale heterogeneity test.**

| urban scale | (1) | (2) | (3) |
|---|---|---|---|
| | **big city** | **medium city** | **small city** |
| Explained variable | lnEmp | lnEmp | lnEmp |
| treat×Post_{12} | 0.0230** | 0.0185*** | 0.0119 |
| | (0.00895) | (0.00693) | (0.0283) |
| lnagglomeration | 0.0918* | 0.0405* | 0.133 |
| | (0.0489) | (0.0214) | (0.0946) |
| lnconsumption | 0.00576 | 0.0158 | 0.0169 |
| | (0.0198) | (0.0108) | (0.0355) |
| lnwage | 0.189*** | 0.101*** | 0.0815* |
| | (0.0454) | (0.0383) | (0.0457) |
| lnsupply | 0.183*** | 0.119*** | 0.162*** |
| | (0.0185) | (0.0137) | (0.0187) |
| Constant | 2.997*** | 2.929*** | 2.091*** |
| | (0.528) | (0.407) | (0.662) |
| Observations | 1,681 | 1,375 | 1,218 |
| R-squared | 0.102 | 0.142 | 0.100 |
| Number of urban | 163 | 173 | 159 |

Note:

***, **, and * denote significance at the 1%, 5%, and 10% levels, respectively. Standard errors are in parentheses.

Therefore, the productivity improvement effect generated by expanding imports has a more significant promoting effect on manufacturing employment in large and medium-sized cities.

**6.4.2 Heterogeneity of urban location.** We divide cities into three categories: eastern, central, and western cities. We then estimate the impact of expanding imports on the manufacturing employment of cities in different locations. Table 7 shows that regardless of location, expanding imports effectively promotes urban employment. However, the heterogeneity of urban locations creates differences in the degree to which expanding imports enhance urban employment. Specifically, expanding imports has the greatest impact on the employment of eastern cities, followed by that of western cities, and it has the smallest impact in central cities.

The main reason is that since the reform and opening up, the three major agglomerations (manufacturing agglomeration, foreign trade agglomeration, and FDI agglomeration) have been prominent in the eastern coastal areas of China, and the productivity improvement effect and product quality upgrading effect resulting from expanding imports have a higher promoting effect on the employment level of eastern cities than on central and western cities. In addition, the One Belt and One Road initiative is one of the important measures for accelerating China's opening up and building a new pattern of comprehensive opening up. Compared with eastern and central cities, an increase in trade between China and countries along the One Belt and One Road can produce a more significant employment spillover effect on western cities, thus promoting the growth of manufacturing employment in these cities.

## 7 Conclusion and policy implications

### 7.1 Conclusion

We study the effect of expanding imports on urban manufacturing employment based on matched data from the China Urban Statistical Yearbook and China Regional Economic

**Table 7. Urban location heterogeneity test.**

| Urban location | (1) | (2) | (3) |
|---|---|---|---|
| | **Eastern city** | **Middle city** | **West city** |
| Explained variable | lnEmp | lnEmp | lnEmp |
| treat×Post$_{12}$ | 0.112*** | 0.0320** | 0.0754*** |
| | (0.0178) | (0.0143) | (0.0259) |
| lnagglomeration | 0.0805*** | 0.250*** | 0.164 |
| | (0.0290) | (0.0613) | (0.222) |
| lnconsumption | 0.217*** | 0.00917 | 0.00663 |
| | (0.0123) | (0.0199) | (0.0203) |
| lnwage | 0.254*** | 0.0218 | 0.136*** |
| | (0.0372) | (0.0370) | (0.0492) |
| lnsupply | 0.149*** | 0.135*** | 0.171*** |
| | (0.0105) | (0.0127) | (0.0203) |
| Constant | 3.877*** | 2.290*** | 2.606*** |
| | (0.402) | (0.450) | (0.582) |
| Observations | 1,515 | 1,633 | 1,127 |
| R-squared | 0.217 | 0.115 | 0.105 |
| Number of urban | 101 | 109 | 75 |

Note:

\*\*\*, \*\*, and \* denote significance at the 1%, 5%, and 10% levels, respectively. Standard errors are in parentheses.

Statistical Yearbook during 2005–2019. To address the endogeneity issue between expanding imports and urban employment, we use the *Guiding Opinions* issued by the State Council in 2012 as a natural experiment to study the effect and mechanism by which expanding imports impacts urban manufacturing employment. We find that: (1) Expanding imports has a significant positive impact on urban manufacturing employment. This conclusion is still robust after a series of robustness tests. (2) Mechanism testing shows that improving productivity and upgrading product quality are important mechanisms through which expanding imports promote urban manufacturing employment. (3) The results of the heterogeneity analysis indicate that the impact of expanding imports on urban manufacturing employment varies depending on the location and size of cities. Specifically, the impact of expanding imports on manufacturing employment in large and medium-sized cities is significantly positive but is not significant in small cities. Expanding imports has effectively promoted manufacturing employment in cities in different locations. However, the heterogeneity of urban location results in differences in the degree of the improvements: expanding imports has the greatest effect on manufacturing employment in eastern cities, followed by western cities and then central cities.

## 7.2 Policy implications

Our findings confirm that expanding imports can increase urban manufacturing employment, which has important policy implications. Developing countries continue to implement expanding import strategies, such as promoting the facilitation of import trade, lowering technology trade barriers, and motivating the mutual recognition of environmental protection and green standards. Fully leveraging the mechanism of expanding imports enhances employment, importing intermediates that facilitate technological progress and upgrades to product quality. However, the government also needs to be aware of the heterogeneous impacts of enlarging imports on employment and implement differentiated import expansion policies. The government needs to further increase policy support for small and central cities and encourage them to actively expand imports. Although expanding imports have increased the employment scale of eastern and western cities, as well as large and medium-sized cities, these cities also need to prevent the adverse effects on local employment from increased foreign competition brought by expanding imports.

## 7.3 Suggestions for future research

Future research can further reveal the impact of expanding imports on urban employment by distinguishing the source of imports and the types of imported products. In terms of heterogeneity analysis, the impact of imports from different industries on urban employment can be studied by distinguishing manufacturing industries.

## Supporting information

**S1 Data.**
(DTA)

**S2 Data.**
(DTA)

**S3 Data.**
(DTA)

## Acknowledgments

The authors wish to thank anonymous referees for all value comments. The author is responsible for any remaining errors.

## Author Contributions

**Conceptualization:** Shiping Wang, Chunyan Zhao.

**Data curation:** Shiping Wang, Chunyan Zhao.

**Formal analysis:** Shiping Wang, Chunyan Zhao.

**Funding acquisition:** Shiping Wang, Chunyan Zhao.

**Investigation:** Shiping Wang, Chunyan Zhao.

**Methodology:** Shiping Wang, Chunyan Zhao.

**Project administration:** Shiping Wang, Chunyan Zhao.

**Resources:** Shiping Wang, Chunyan Zhao.

**Software:** Shiping Wang, Chunyan Zhao.

**Supervision:** Shiping Wang, Chunyan Zhao.

**Validation:** Shiping Wang, Chunyan Zhao.

**Visualization:** Shiping Wang, Chunyan Zhao.

**Writing – original draft:** Shiping Wang, Chunyan Zhao.

**Writing – review & editing:** Shiping Wang, Chunyan Zhao.

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
