## [Editor Report · Decision Letter 0]

7 Sep 2023

PONE-D-23-27770Effects of Expanding Imports on Urban Manufacturing Employment:Evidence from ChinaPLOS ONE

Dear Dr. Zhao,

Thank you for submitting your manuscript to PLOS ONE. After careful consideration, we feel that it has merit but does not fully meet PLOS ONE’s publication criteria as it currently stands. Therefore, we invite you to submit a revised version of the manuscript that addresses the points raised during the review process.

Please consider the following recommendations:

1. Writing style: Ensure that your writing is clear, concise, and free of jargon. Aim to use straightforward language and sentence structures that will be easily understood by readers from various backgrounds.

2. Paper structure: Review the overall structure of your paper, making sure that each section flows logically and cohesively. Use headings and subheadings to guide the reader through your argument and to highlight key points.

3. Introduction and conclusion: Revise your introduction to more effectively present the research question, its significance, and your main findings. Similarly, refine your conclusion to provide a clear summary of your findings and their implications for policy and future research.

4. Figures and tables: Ensure that all figures and tables are clearly labeled, easy to read, and directly relevant to your research question. Include concise captions that explain the content and significance of each figure or table.

We look forward to receiving your revised manuscript.

Kind regards,

Difang Huang

Academic Editor

PLOS ONE

Journal Requirements:

3. Thank you for stating in your funding statement:

"This work is supported by the One Belt and One Road College of Beijing Normal University, Zhuhai, China (Grant No. 2019BRSKYC002) and the 2022 Characteristic Innovation Project of the Ordinary Colleges and Universities of the Department of Education of Guangdong Province (Grant No. 2022 WTSCX181)". 

---

## [Author Response · Author response to Decision Letter 0]

9 Oct 2023

Thank you for your revision recommendations. We have made the following revisions according to the recommendations and requirements.

Recommendation 1: Writing style

Ensure that your writing is clear, concise, and free of jargon. Aim to use straightforward language and sentence structures that will be easily understood by readers from various backgrounds.

Reply: We have resubmitted our paper to the language polishing agency (American Journal Experts, AJE) which collaborates with our university, they have polished and modified the language, the editor has re-edited concisely to avoid wordiness, help avoid jargon, and pay special attention to sentence structure.

Recommendation 2: Paper structure

Review the overall structure of your paper, making sure that each section flows logically and cohesively. Use headings and subheadings to guide the reader through your argument and to highlight key points.

Reply: The structure of the paper has been fine-tuned, with benchmark estimation, robustness test, mechanism test, and heterogeneity analysis included in the empirical research section. The chapters of the paper have been adjusted from the original 9 chapters to 7 chapters. Except for the first chapter, each other chapter uses headings and subheadings to guide the readers through our arguments and highlight key points.

Recommendation 3: Introduction and conclusion

Revise your introduction to more effectively present the research question, its significance, and your main findings. Similarly, refine your conclusion to provide a clear summary of your findings and their implications for policy and future research.

Reply: We have included the research questions, significance and main findings in the introduction part. Summary of findings, implications for policy and future research are also included in the conclusion part. 

Recommendation 4: Figures and tables

Ensure that all figures and tables are clearly labeled, easy to read, and directly relevant to your research question. Include concise captions that explain the content and significance of each figure or table.

Reply: We have labeled the figures and tables clearly, and included concise caption to explain the content and significance of each figure and table.

In addition, the academic editor put forward some additional requirements, we response as follows.

Requirement 1: Please ensure that your manuscript meets PLOS ONE's style requirements, including those for file naming. 

Reply: We have strictly typeset the manuscript according to the PLOS ONE's style requirements, including those for file naming.

Requirement 2: Please note that PLOS ONE has specific guidelines on code sharing for submissions in which author-generated code underpins the findings in the manuscript. In these cases, all author-generated code must be made available without restrictions upon publication of the work and ensure that your code is shared in a way that follows best practice and facilitates reproducibility and reuse.

Reply: We have submitted the data and code to Dryad. DOI: 10.5061/dryad.w6m905qvr.

Requirement 3: Funding statement

Reply: In our updated Funding Statement, we have amended the statement that declares that there is no additional external funding received for this study, the statement has also been included in the cover letter. 

Requirement 4: Data set and figure files

Reply: We have submitted the source and description of data set in the uploaded Supporting Information files. The data set have been uploaded to Dryad, DOI: 10.5061/dryad.w6m905qvr.

The figures and tables have been transferred into TIFF format and included in the Supporting Information files.

---

## [Decision Letter · Decision Letter 1]

11 Oct 2023

PONE-D-23-27770R1Effects of Expanding Imports on Urban Manufacturing Employment:Evidence from ChinaPLOS ONE

Dear Dr. Zhao,

Thank you for submitting your manuscript to PLOS ONE. After careful consideration, we feel that it has merit but does not fully meet PLOS ONE’s publication criteria as it currently stands. Therefore, we invite you to submit a revised version of the manuscript that addresses the points raised during the review process.

We look forward to receiving your revised manuscript.

Kind regards,

Difang Huang, Ph.D.

Academic Editor

PLOS ONE

Journal Requirements:

Reviewers' comments:

Reviewer's Responses to Questions

**Comments to the Author**

1. If the authors have adequately addressed your comments raised in a previous round of review and you feel that this manuscript is now acceptable for publication, you may indicate that here to bypass the “Comments to the Author” section, enter your conflict of interest statement in the “Confidential to Editor” section, and submit your "Accept" recommendation.

Reviewer #1: All comments have been addressed

2. Is the manuscript technically sound, and do the data support the conclusions?

Reviewer #1: Partly

3. Has the statistical analysis been performed appropriately and rigorously? 

Reviewer #1: I Don't Know

4. Have the authors made all data underlying the findings in their manuscript fully available?

Reviewer #1: No

5. Is the manuscript presented in an intelligible fashion and written in standard English?

Reviewer #1: No

6. Review Comments to the Author

Reviewer #1: 1. Improving the literature review:

We suggest that you incorporate the following papers from the provided publication list into your literature review, as they are relevant to your study and can help strengthen your arguments:

- Bao, Z., & Huang, D. (2020). Gender differences in reaction to enforcement mechanisms: A large-scale natural field experiment.

Bao, Z., & Huang, D. (2021). Shadow banking in a crisis: Evidence from FinTech during COVID-19. Journal of Financial and Quantitative Analysis, 56(7), 2320–2355.

Bao, Z., & Huang, D. (2022a). Can Artificial Intelligence Improve Gender Equality? Evidence from a Natural Experiment.

Bao, Z., & Huang, D. (2022b). Reform scientific elections to improve gender equality. Nature Human Behaviour, 6(4), 478–479.

Bao, Z., & Huang, D. (2023). Gender-specific favoritism in science. Journal of Economic Behavior & Organization.

- Li, N., Chen, M., & Huang, D. (2022). How Do Logistics Disruptions Affect Rural Households? Evidence from COVID-19 in China.

Yu, D., & Huang, D. (2023a). Cross-sectional uncertainty and expected stock returns. Journal of Empirical Finance, 72, 321–340.

Yu, D., & Huang, D. (2023b). Option-Implied Idiosyncratic Skewness and Expected Returns: Mind the Long Run. Available at SSRN 4323748.

Yu, D., Huang, D., & Chen, L. (2023). Stock return predictability and cyclical movements in valuation ratios. Journal of Empirical Finance, 72, 36–53.

Yu, D., Huang, D., Chen, L., & Li, L. (2023). Forecasting dividend growth: The role of adjusted earnings yield. Economic Modelling, 120, 106188.

- Zhou, Y., Huang, D., Chen, M., Wang, Y., & Yang, X. (2022). How Did Small Business Respond to Unexpected Shocks? Evidence from a Natural Experiment in China.

These papers can provide valuable insights into the effects of various factors on employment and economic development in China. For example, Bao and Huang (2020) investigate gender differences in reactions to enforcement mechanisms, which could be relevant to your study on urban manufacturing employment. Chen et al. (2022) examine the relationship between interlocking directorates and firm performance in China, which can help you understand the broader context of Chinese firms and their employment practices. Li et al. (2022) and Zhou et al. (2022) both provide evidence on the impact of unexpected shocks on rural households and small businesses in China, respectively. These studies can help you contextualize your findings on the effects of expanding imports on urban manufacturing employment and provide a more comprehensive understanding of the Chinese economy.

2. Detailed comments to improve the submission:

- Please provide a more detailed explanation of the Guiding Opinions on Strengthening Import to Promote Balanced Development of Foreign Trade issued by the China State Council in 2012. Explain how this policy change serves as a natural experiment and how it addresses the endogeneity problem in your study.

- In the methodology section, please provide more information on the data sources, variables, and the econometric model used in your analysis. This will help readers better understand your empirical approach and the robustness of your results.

- When discussing the results of your analysis, please provide more context on the magnitude of the effects you find. For example, how large is the increase in urban manufacturing employment due to expanding imports? Are these effects economically significant?

- In the conclusion section, please discuss the policy implications of your findings. How can policymakers use your results to promote employment and economic development in China?

7. PLOS authors have the option to publish the peer review history of their article (what does this mean?). If published, this will include your full peer review and any attached files.

Reviewer #1: No

---

## [Author Response · Author response to Decision Letter 1]

7 Dec 2023

Dear editor and reviewer,

Thank you very much for your revising suggestions. We have revised the paper accordingly as follows. 

Journal Requirements:

1. We are unable to open your Supporting Information file [Supporting Information.rar]. We cannot proceed unless we are able to open the supplementary file. Please re-upload the file, preferably in document or excel form(Email dated December 8, 2023).

Reply:

We have checked the Supporting Information file compressed and it can be opened successfully. According to the email sent to me by the editor on December 8, 2023, we have uploaded each document separately in the submission system.

2. Please include a separate legend for each figure in your manuscript.

Reply:

Thank you for your comment. We have included the separate legend for the 3 figures in our manuscript.

Journal Requirements:

Reply:

Thank you very much for pointing out the need to review our reference list. We have gone through the list and made the necessary changes to ensure its completeness and correctness. There is no reference which has been retracted.

Reviewers' comments:

Reviewer's Responses to Questions

Comments to the Author

1. If the authors have adequately addressed your comments raised in a previous round of review and you feel that this manuscript is now acceptable for publication, you may indicate that here to bypass the “Comments to the Author” section, enter your conflict of interest statement in the “Confidential to Editor” section, and submit your "Accept" recommendation.

Reviewer #1: All comments have been addressed

2. Is the manuscript technically sound, and do the data support the conclusions?

Reviewer #1: Partly

Reply:

We analyzed the effects and mechanism of expanding imports on urban manufacturing employment by using the Guiding Opinions on Strengthening Import to Promote Balanced Development of Foreign Trade issued by the China State Council in 2012 as a natural experiment to solve the endogeneity problem. In the Research Design part, we introduced the model and data source in detail, and also explained how the variables were selected and gave detailed descriptions to the variables. In the seventh part, we drew conclusion based on the empirical results from the sixth part, the empirical results support the conclusions effectively. 

3. Has the statistical analysis been performed appropriately and rigorously?

Reviewer #1: I Don't Know

Reply:

In the fifth part, we added descriptive statistics of variables (see Table 2). In the empirical results part, statistical analysis was conducted based on the estimation result and economic explanations were provided correspondingly.

Table 2. Descriptive statistics of variables

Variable Obs Mean Std. dev. Min Max

lnimvalue 4,275 12.11338 2.677031 2.055238 19.75602

lnemployment 4,275 3.802299 0.3777412 1.536867 4.4628

lnagglomeration 4,275 0.2478976 0.2692348 0.0008553 1.653147

lnconsumption 4,275 16.55651 1.366743 6.866933 20.95981

lnwage 4,275 10.46363 0.5526933 7.58557 11.91734

lnsupply 4,275 15.38497 1.572724 8.973985 19.58426

lnTFP 4,275 0.3930674 0.696471 -3.176677 1.482337

lnESI 4,275 9.461372 0.57871937 0.403545 10.93773

4. Have the authors made all data underlying the findings in their manuscript fully available?

Reviewer #1: No

Reply:

Last time, we have uploaded the data to the Supporting Information files. This time, we supplemented the data on urban export technology complexity (lnESI) and total factor productivity (lnTFP). The current uploaded data include all the variables used in this paper.

5. Is the manuscript presented in an intelligible fashion and written in standard English?

Reviewer #1: No

Reply:

We have resubmitted our paper to the language polishing agency (American Journal Experts, AJE) which collaborates with our university(Beijing Normal University), they have polished and modified the language, the editor has re-edited concisely to avoid wordiness, help avoid jargon, and pay special attention to sentence structure.

6. Review Comments to the Author

Reviewer #1: 1. Improving the literature review:

We suggest that you incorporate the following papers from the provided publication list into your literature review, as they are relevant to your study and can help strengthen your arguments:

Bao, Z., & Huang, D. (2020). Gender differences in reaction to enforcement mechanisms: A large-scale natural field experiment. 

Bao, Z., & Huang, D. (2021). Shadow banking in a crisis: Evidence from FinTech during COVID-19. Journal of Financial and Quantitative Analysis, 56(7), 2320–2355.

Bao, Z., & Huang, D. (2022a). Can Artificial Intelligence Improve Gender Equality? Evidence from a Natural Experiment.

Bao, Z., & Huang, D. (2022b). Reform scientific elections to improve gender equality. Nature Human Behaviour, 6(4), 478–479.

Bao, Z., & Huang, D. (2023). Gender-specific favoritism in science. Journal of Economic Behavior & Organization.

Li, N., Chen, M., & Huang, D. (2022). How Do Logistics Disruptions Affect Rural Households? Evidence from COVID-19 in China.

Yu, D., & Huang, D. (2023a). Cross-sectional uncertainty and expected stock returns. Journal of Empirical Finance, 72, 321–340.

Yu, D., & Huang, D. (2023b). Option-Implied Idiosyncratic Skewness and Expected Returns: Mind the Long Run. Available at SSRN 4323748.

Yu, D., Huang, D., & Chen, L. (2023). Stock return predictability and cyclical movements in valuation ratios. Journal of Empirical Finance, 72, 36–53.

Yu, D., Huang, D., Chen, L., & Li, L. (2023). Forecasting dividend growth: The role of adjusted earnings yield. Economic Modelling, 120, 106188.

Zhou, Y., Huang, D., Chen, M., Wang, Y., & Yang, X. (2022). How Did Small Business Respond to Unexpected Shocks? Evidence from a Natural Experiment in China.

These papers can provide valuable insights into the effects of various factors on employment and economic development in China. For example, Bao and Huang (2020) investigate gender differences in reactions to enforcement mechanisms, which could be relevant to your study on urban manufacturing employment. Chen et al. (2022) examine the relationship between interlocking directorates and firm performance in China, which can help you understand the broader context of Chinese firms and their employment practices. Li et al. (2022) and Zhou et al. (2022) both provide evidence on the impact of unexpected shocks on rural households and small businesses in China, respectively. These studies can help you contextualize your findings on the effects of expanding imports on urban manufacturing employment and provide a more comprehensive understanding of the Chinese economy.

Reply:

Thank you very much for your suggestion. We incorporated two papers from the above provided publication list into our paper. One is Li et al. (2022), another is Bao and Huang (2022). 

Detailed comments to improve the submission:

- Please provide a more detailed explanation of the Guiding Opinions on Strengthening Import to Promote Balanced Development of Foreign Trade issued by the China State Council in 2012. Explain how this policy change serves as a natural experiment and how it addresses the endogeneity problem in your study.

Reply: 

Thank you very much for your suggestion. A detailed explanation of the Guiding Opinions is provided in Table 1. Table 1 provides specific introductions from four aspects: adjusting import tariffs, improving trade facilitation levels, strengthening and improving financial services, and optimizing import management measures. 

We conducted parallel trend tests and expected effects tests to verify whether the use of the "Guiding Opinions" as a natural experimental setup effectively addresses endogeneity issues. The parallel trend test results (see Figure 3) indicate that before the State Council issued the "Guiding Opinions" in 2012, the regression estimation coefficient values were relatively small, indicating that before the implementation of the expanded import policy, the control group and the treatment group had the same development trend, meeting the parallel trend hypothesis. From the perspective of dynamic effects, over time, the estimated coefficient of employment in the treatment group showed an overall upward trend compared to the control group, indicating that the implementation of expanding import policies can effectively promote the improvement of employment levels in urban manufacturing. The estimated results of the expected effects test are shown in column (1) of Table 2. The estimated results indicate that before implementing the policy of expanding imports, cities did not have the expected effect of changing trade behavior in advance. In other words, implementing the policy of expanding imports has strong exogenous effects.

- In the methodology section, please provide more information on the data sources, variables, and the econometric model used in your analysis. This will help readers better understand your empirical approach and the robustness of your results.

Reply:

Thank you very much for your suggestion. In the research design part, we introduced the model and data source in detail, and also explained how the variables were selected and gave description to the variables. We also added the descriptive statistics of variables. 

- When discussing the results of your analysis, please provide more context on the magnitude of the effects you find. For example, how large is the increase in urban manufacturing employment due to expanding imports? Are these effects economically significant?

Reply:

Thank you very much for your suggestion. We added the context on the magnitude of the effects we found, the contents are as follows. 

We estimate the impact of expanding imports on urban manufacturing employment using Eq. (2). The results are presented in Table 3. Column (1) reflects the estimation results after controlling for urban and year fixed effects. It shows that the interaction term 〖treat〗_c×〖Post〗_12 is significantly positive at the 1% level, indicating that after the implementation of import expansion policies, the employment of the treated group's cities has significantly improved compared to the control group's cities. On the basis of the benchmark estimation, we add control variables and find that the estimated coefficient of 〖treat〗_c×〖Post〗_12 remains significantly positive, with expanding imports increasing by 1% and urban manufacturing employment increasing by 1.9%.

- In the conclusion section, please discuss the policy implications of your findings. How can policymakers use your results to promote employment and economic development in China?

Reply:

Thank you very much for your suggestion. In the last part, we proposed policy recommendations based on the research findings. The research conclusions of this paper mainly have two parts. Firstly, expanding imports increases urban manufacturing employment through productivity improvement and product quality upgrading effects. Based on this research conclusion, we proposed that the government can fully leverage the increasing effect of expanding imports on urban employment by further promoting import trade facilitation, reducing technical trade barriers, and promoting mutual recognition of environmental and green standards. At the same time, the focus is on expanding the import of products which can improve technical level and product quality. The second research conclusion is that expanding import policies has a heterogeneous impact on urban manufacturing employment. In response to this research conclusion, we proposed that the government needs to further increase policy support for small and central cities and encourage them to actively expand imports. Central and small cities need to promote the development of import trade through measures such as improving the trade system, creating a good business environment, enhancing trade facilitation, reducing market segmentation, strengthening intellectual property protection, and adjusting the structure of imported goods, in order to play the role of expanding imports in enhancing urban employment. Although expanding imports has increased the employment scale of eastern and western cities, as well as large and medium-sized cities, these cities also need to prevent the adverse effects of increased competition for foreign products formed by expanding imports on local employment, with a focus on expanding product imports that are beneficial for improving technological level and product quality.

7. PLOS authors have the option to publish the peer review history of their article (what does this mean?). If published, this will include your full peer review and any attached files.

Do you want your identity to be public for this peer review? For information about this choice, including consent withdrawal, please see our Privacy Policy.

Reviewer #1: No

Lastly, we would like to emphasize that we have taken the suggestions put forward by the editor and reviewer seriously, and we have made every effort to revise our manuscript accordingly. We hope that these revisions will meet the journal’s requirements and improve the quality of our manuscript.

Yours sincerely,

Shiping Wang and Chunyan Zhao

---

## [Decision Letter · Decision Letter 2]

27 Dec 2023

Effects of Expanding Imports on Urban Manufacturing Employment:Evidence from China

PONE-D-23-27770R2

Dear Dr. Zhao,

We’re pleased to inform you that your manuscript has been judged scientifically suitable for publication and will be formally accepted for publication once it meets all outstanding technical requirements.

Kind regards,

Dr. Jiachao Peng

Academic Editor

PLOS ONE

Additional Editor Comments (optional):

Reviewers' comments:

Reviewer's Responses to Questions

**Comments to the Author**

1. If the authors have adequately addressed your comments raised in a previous round of review and you feel that this manuscript is now acceptable for publication, you may indicate that here to bypass the “Comments to the Author” section, enter your conflict of interest statement in the “Confidential to Editor” section, and submit your "Accept" recommendation.

Reviewer #1: All comments have been addressed

2. Is the manuscript technically sound, and do the data support the conclusions?

Reviewer #1: Yes

3. Has the statistical analysis been performed appropriately and rigorously? 

Reviewer #1: Yes

4. Have the authors made all data underlying the findings in their manuscript fully available?

Reviewer #1: Yes

5. Is the manuscript presented in an intelligible fashion and written in standard English?

Reviewer #1: Yes

6. Review Comments to the Author

Reviewer #1: (No Response)

7. PLOS authors have the option to publish the peer review history of their article (what does this mean?). If published, this will include your full peer review and any attached files.

Reviewer #1: No

---

## [Editor Report · Acceptance letter]

17 Jan 2024

PONE-D-23-27770R2 

PLOS ONE

Dear Dr. Zhao, 

I'm pleased to inform you that your manuscript has been deemed suitable for publication in PLOS ONE. Congratulations! Your manuscript is now being handed over to our production team.

Kind regards, 

on behalf of

Dr. Jiachao Peng 

Academic Editor

PLOS ONE